# Goethite Nanorods: Synthesis and Investigation of the Size Effect on Their Orientation within a Magnetic Field by SAXS

**DOI:** 10.3390/nano10122526

**Published:** 2020-12-16

**Authors:** Stephan Hinrichs, Larissa Grossmann, Eike Clasen, Hannah Grotian genannt Klages, Dieter Skroblin, Christian Gollwitzer, Andreas Meyer, Birgit Hankiewicz

**Affiliations:** 1Institute of Physical Chemistry, University of Hamburg, Grindelallee 117, 20146 Hamburg, Germany; stephanhinrichs64@gmail.com (S.H.); fcux268@studium.uni-hamburg.de (L.G.); eike.clasen@gmx.de (E.C.); hannah.grotian@chemie.uni-hamburg.de (H.G.g.K.); andreas.meyer@chemie.uni-hamburg.de (A.M.); 2Physikalisch-Technische Bundesanstalt (PTB), Abbestraße 2-12, 10587 Berlin, Germany; dieter.skroblin@ptb.de (D.S.); christian.gollwitzer@ptb.de (C.G.)

**Keywords:** goethite, akaganeite, anisotropy, hydrothermal, orientation, particle synthesis

## Abstract

Goethite is a naturally anisotropic, antiferromagnetic iron oxide. Following its atomic structure, crystals grow into a fine needle shape that has interesting properties in a magnetic field. The needles align parallel to weak magnetic fields and perpendicular when subjected to high fields. We synthesized goethite nanorods with lengths between 200 nm and 650 nm in a two-step process. In a first step we synthesized precursor particles made of akaganeite (β-FeOOH) rods from iron(III)chloride. The precursors were then treated in a hydrothermal reactor under alkaline conditions with NaOH and polyvinylpyrrolidone (PVP) to form goethite needles. The aspect ratio was tunable between 8 and 15, based on the conditions during hydrothermal treatment. The orientation of these particles in a magnetic field was investigated by small angle X-ray scattering (SAXS). We observed that the field strength required to trigger a reorientation is dependent on the length and aspect ratio of the particles and could be shifted from 85 mT for the small particles to about 147 mT for the large particles. These particles could provide highly interesting magnetic properties to nanocomposites, that could then be used for sensing applications or membranes.

## 1. Introduction

Iron oxides offer a lot of different phases with widely varying shapes and magnetic properties. While the bulk of the research focuses on the ferro- and ferrimagnetic properties of pure iron, magnetite or maghemite, there are also the antiferromagnetic species that show interesting magnetic properties. One example of these is goethite. Goethite was first used as a pigment by prehistoric painters and is still used for the same purpose [1]. In the meantime it has also found new applications in the area of heavy metal absorption for the treatment of wastewater [2,3,4]. Although the bulk material shows antiferromagnetic properties, nanoparticles add an interesting new spin, because of their increased surface area. Its extraordinary properties were first observed by Lemaire and coworkers [5,6,7,8]. They showed that goethite forms a liquid crystalline phase above 8.5 wt.%, simply due to its elongated particle shape in the nanometer regime. If the concentration is below that threshold, the orientation of the ensemble is solely dependent on the applied magnetic field [9]. Goethite aligns parallel to weak magnetic fields while orienting perpendicular to the field at high field strengths. This behavior is unique to nanoparticles, as the effect is attributed to the surface moments of the particles that are free to orient themselves in response to a magnetic field. 

Understanding this behavior and how it is connected to the shape of the particles holds important insights into the properties of goethite and into nanoparticles in general. Therefore, we conducted experiments to investigate how to tailor the synthesis of goethite nanoparticles. We wanted to gain control over the length of the particles formed in order to see whether this has an influence on the magnetic properties. To the best of our knowledge, this has not been done with success before. However, there have been studies on branched goethite particles [10], very large goethite rods [11], goethite wires [12], spherical nano goethite [13] and on the influence of different surfactants [14]. Different from most synthesis protocols which rely on the aging of a precursor solution for several days [6,10,15], the method described here takes only four hours. Our synthesis follows a two-step protocol: first, the synthesis of a β-FeOOH precursor, and second, a hydrothermal treatment step guided by poly vinyl pyrrolidone (PVP) at pH 12 to convert the β-FeOOH into α-FeOOH. The particles are stabilized by PVP so we used a sterical barrier to counteract agglomeration and precipitation. As shown previously, a sterical barrier has no detrimental influence when compared to charge-based stabilization [16]. The alignment of the particles was determined by Small Angle X-ray Scattering (SAXS). We were able to show that the length of the nanoparticles influences the magnetic field required for reorientation of the goethite rods.

## 2. Materials and Methods 

FeCl_3_·6H_2_O (≥97%) was obtained from Sigma-Aldrich (Darmstadt, Germany), Na_2_HPO_4_ (≥99.5%) from Merck (Darmstadt, Germany), NaOH (≥99%) from Grüssing (Filsum, Germany) and PVP K-15 average M_w_ = 10,000 Da was obtained from TCI Chemicals (Zwijndrecht, Belgium), all chemicals were used as received, without further purification. 

For the reactions in a hydrothermal reactor we used a system made by Berghof instruments (Eningen unter Achalm, Germany). The hydrothermal reactor consists of a Teflon inlet with an internal volume of 25 mL, which is placed into a stainless-steel autoclave. The system was pressurized by nitrogen with a pressure of 10 bar. During the reaction at 160 °C the pressure peaked at 15 bar. 

Transmission electron micrographs (TE micrograph) were recorded on a FEI Tecnai G2 spirit twin at an accelerating voltage of 120 kV. For sample preparation, a drop of diluted particle suspension was deposited on a carbon-coated copper grid. The solvent was removed by filter paper. The particle size was evaluated by measuring the length and width of 100 particles in each sample using the freeware software ImageJ (Version 1.50i, National institues of health, USA) [17]. 

X-ray diffraction (XRD) was done on a Philips X’Pert PRO MPD machine (Almelo, The Netherlands) with an X-ray wavelength of 1.54 Å. For the evaluation of the crystal phases present, we used the software Highscore X’pert PRO by PanAnalytical (Version 2.2.3, Almelo, The Netherlands) with the JCPDS PDF Nr. 01-081-0462 for goethite and PDF Nr. 01-075-1594 for akaganeite. For sample preparation, after washing to neutral pH, ca. 100 µL of the particle dispersion was deposited on a silicon wafer and dried in air. 

Inhouse SAXS data were taken on an Incoatec X-ray source IµS with Quazar Monetl optics (Geesthacht, Germany). The focal spot diameter was 700 µm, the wavelength was 0.154 nm, the detector distance was 1.6 m and the detector was a Rayonix SX165 CCD-Detector (Evanston, IL, USA). Samples were prepared as colloidal suspensions. Particle content was adjusted to 2.5 wt.% (m/m) by centrifugation and addition of water. The magnetic field was established by two cubic permanent magnets mounted on two sleds that can be moved symmetrically by a shaft. The distance between the magnets determined the magnetic field strength and was reduced from 50 mm in 1 mm steps. The minimum field strength with this setup at 50 mm is 35 mT, the highest field strength employed in this work was 533 mT at an intermagnet distance of 4.5 mm. Baseline images without a magnetic field were taken in a standard capillary setup. For each image, two frames were taken, each with a measurement time of 300 s, so 10 min measurement time for each field strength. 

In addition, SAXS experiments were conducted at the four-crystal monochromator (FCM) beamline of the PTB laboratory located at the BESSY II storage ring [18]. The scattering pattern was collected on a vacuum-compatible version of the PILATUS 1M detector with a pixel size of 172 × 172 µm^2^ at a distance of 4.527 m from the sample [19]. The photon energy was set to 8 keV. Images were taken with an integration time of 60 s. To apply a magnetic field, a similar setup with permanent magnets as described above was used, with the difference that the magnets were mounted into a PEEK frame, thus only allowing a distinct set of field strengths, and thereby limiting the total number of adjustable field values. An azimuthal profile of the recorded intensity distribution was taken and evaluated using the DPDAK software (Version 1.3.1., Hamburg, Germany) [20]. This yields an angular profile of the integrated intensity along each angle with a periodic modulation. The intensity of this modulation indicates the degree of alignment in the sample. The direction of orientation can be estimated from the position of the peak at 360° (180°/0°). We defined the angle by using the direction of the magnetic field as 0°. We used a Lorentzian fit on the averaged azimuthal intensity to get the angle for the maximum position of the fit. To describe the direction of the orientation we used the nematic order parameter, the Herman orientation function:(1)$$f_{H}=\frac{1}{2}\left(3\langle \cos^{2}\phi \rangle -1\right)$$

Here ϕ is the angle between the field direction and the long axis of the rods, and *f_H_* is the value of the Herman orientation function. It has a range from −0.5, which means a perpendicular orientation of the axis to the magnetic field over 0, and thus none or isotropic orientation to 1, which means a parallel orientation of the long particle axis to the field direction [21]. Since the main feature observed in our SAXS setup is the short axis of the nanorods, the orientation was determined by taking the maximum position of the fit and adding 90° to accurately describe the direction of the long axis. We determined the Herman orientation function for different particle sizes at different magnetic field strengths.

## 3. Synthesis Protocol

The synthesis of goethite nanorods was divided into two parts: the synthesis of the precursor by hydrolysis of Fe(III)Cl_3_ salts and the transformation of the *β*-FeOOH precursor into goethite nanorods in a hydrothermal reactor. The first step is done in a 500 mL single-neck flask. 2.176 mL of a 7.8 mg/mL Na_2_HPO_4_ solution (0.095 mmol) in water are added to 240 mL of demineralized water. 1.3170 g of FeCl_3_·6H_2_O (4.87 mmol) are added to the solution and heated to 80 °C. The solution is maintained at this temperature and stirred with a magnetic stirring bar at 500 rpm for two days. After about one hour of reaction time, the particle solution changes color from the orange color of Fe(III)Cl_3_ to a red/brown associated with akaganeite. The weighing is done to ensure that the final precursor solution is at a concentration of 0.1 mol/L or a mass fraction of 0.8885 wt.% (m/m). For one of the syntheses, the precursor particles have been dialyzed prior to the second synthesis step to remove leftover educts (leads to goethite sample G_D). 

In the second step, the β-FeOOH precursors are treated in a hydrothermal reactor. At first, 0.5 g (12.5 mmol) of NaOH is added into the Teflon reactor inlet, followed by the addition of 0.1830 g (18.3 µmol) PVP K-15 (sequence 1). Then, 7.354 mL (0.735 mmol) of the precursor solution is added to the reactor and diluted with the same amount of water to yield a concentration of 0.05 mol/L. For large particles the order of the addition was changed so that the precursor was added before the PVP K-15 (sequence 2).

The reactor is equipped with a magnetic stirring bar and sealed. After that, it is set into a heating mantle on a stirring plate, the stirring speed is adapted to the desired particle length (between 100 rpm and 500 rpm). The reactor is pressurized to 10 bars with nitrogen. The solution is stirred for 60 min at room temperature and after that heated to 160 °C with a 60-min temperature ramp. The reaction time for the hydrothermal reaction varied between 2 and 12 h. The specific conditions that led to each product are displayed in Table 1.

The stabilization of the particles in solution was done by adding 0.5 g of PVP. After that the particles were placed in an ultrasonic bath for 5 min and put on a shaker for 2 h. 

## 4. Synthesis: Results and Discussion

The precursor particles were synthesized in a batch size and concentration to be used immediately. The precursor particles had an average length of (42 ± 10) nm and an average width of (10 ± 2) nm and displayed the akaganeite’s typical cigar shape (Figure 1A). The synthesis yielded particles of high purity, and all major peaks in the sample could be assigned to reference peaks of a goethite structure (Figure 1B). After the synthesis, the precursor batch was at a pH value of about 3 and was immediately used in the goethite synthesis. 

It is a well-known fact that nanoparticle synthesis requires much attention to detail in order to be reproducible [22,23], so we investigated some common variations between batches. In a standard protocol (sequence 1), we add the precursor solution to the NaOH and PVP. This leads to short particles around (210 ± 61) nm in length and (27 ± 9) nm in width (G_S) (Figure 2A). Interestingly, the smallest G_S nanorods appear to have the smallest polydispersity of all the samples (Figure 2A, see also Table 2). The reaction was stirred at low speeds and with a small stirring bar, keeping the perturbations minimal. Unsurprisingly the smallest particles were also the most colloidally stable species, during long time experiments. The goethite phase of the samples was confirmed by XRD measurements (Figure 2B).

The intermediate G_M sample with an average rod length of (250 ± 77) nm and width of (27 ± 8) nm shows no very long specimens, but some short rods, that increase the polydispersity (Figure 3A). This configuration was achieved by stirring slowly with a rather big stirring bar for two hours and stabilizing with PVP from the start.

The sequence of addition is of almost as important as the ratio of reactants. PVP given at first into the empty reactor leads to short particles like the ones described above (G_S) (Figure 2A), while PVP added last produces long particles (sequence 2), around (320 ± 97) nm in length and (24 ± 7) nm in width (G_L) (Figure 3B). However, these particles are often accompanied by impurities that appear as much larger particles with a greatly increased width. The impurities were filtered out by careful centrifugation, after the goethite rods had been stabilized by PVP after the reaction was finished. 

The earliest change in terms of chronological order of the synthesis can be induced when the precursor particles are dialyzed against demineralized water prior to the hydrothermal treatment. We dialyzed the precursor for one week, while changing the water daily. By dialyzing the precursor, we wanted to get rid of free Na_2_HPO_4_, but as a side effect we also swapped the chloride ions in the tunnel structure of akaganeite with hydroxide ions [24], which in turn influenced the reaction to goethite (sample G_D). Within the sample there are some very long rods up to a micrometer in length (Figure 4A), but also some large particles which are very broad. In total, the particle size was about (650 ± 330) nm in length and (32 ± 13) nm in width, which was determined via TEM. We assumed these particles to be other iron oxides but could not find strong signals in the XRD (Figure 4B). Since these particles are very similar in size compared to the impurities, it was no longer possible to separate them by centrifugation. This sample was thus omitted from SAXS measurements.

The second investigated parameter, the concentration of the precursor solution, was varied but did not yield any morphological or phase changes (no data shown), so for the sake of reproducibility with the precursor concentration after the synthesis in mind, we adjusted the concentration to 0.05 mol/L. However, the amount of precursor obviously influences the amount of product gained from the reaction. This means that by increasing the precursor concentration, the reaction can be easily scaled up, leading to a higher yield.

The next factor taken into consideration was the addition of PVP to the reaction mixture. The amount of PVP given in the materials section amounts roughly to one molecule of PVP per 60 nm^2^. Increasing the amount of PVP to one molecule per 6 nm^2^ led to marginally longer particles of (236 ± 84) nm (29 ± 11 nm), but with a significantly broader distribution in length as well as in width. This is not further discussed in this publication. Omitting PVP led to short particles measuring (218 ± 70) nm in length and (33 ± 11) nm in width (G_noPVP). However, this included some very small particles that lacked a defined shape. TEM images are not shown here, since sample G_noPVP was very similar to G_S (see Figure 2A).

All samples consist of pure goethite without the common impurities of hematite or other iron oxides. Here we show, as an example, the XRD pattern of sample G_S (Figure 2B). All the major peaks in the sample could be assigned to the goethite reference. Due to the elongation of the particles, almost all particles align to the silicon wafer, leading to the extinction of some crystal reflexes that are present in the bulk goethite reference, while intensifying other reflexes—like the (110) peak, which is very strongly pronounced in our samples. 

The particles counted for the size measurement were chosen at random, not neglecting agglomerates or malformed particles, which lead to a broadening of the size distribution. After the reaction, the particles have a uniform rod-shape. The length of the particles varies between samples, depending on the conditions during the reaction, while the width varies between 20 nm and 30 nm across all observed reactions (Table 2), leading to aspect ratios between 7 and 20. Sample G_L was treated post cleaning by careful centrifugation to isolate the long nanorods and to get rid of large particle impurities. For comparison reasons, the samples are listed with their length and width in aspect ratio in Table 2.

All particles appear polydisperse in length, and the deviation increases with particle length. This is likely due to the random perturbations introduced by prolonged stirring. While some particles grow quite long without colliding with other particles and make up the 500 nm plus species in the TEM images, other particles collide with neighboring particles, and we observe fractures in the nanorods. This is visible in Figure 3B, where some nanorods appear to show rough edges.

Goethite nanorods often agglomerate with their long sides adjacent to each other. This effect, though visible in the images, is minimal, indicating a good dispersion of the nanorods in solution.

The particles are randomly oriented on the TEM grids, with their (100) direction perpendicular to the electron beam, so all images are of the (100) zone of the crystals. The particles show no preferential orientation on the TEM grid, indicating no long-range order or agglomeration. The identical orientation perpendicular to the incident beam is likely caused by the magnetic field employed by the objective lens in the electron microscope, on top of the tendency of elongated objects to align parallel to the sample plane. Some particles appear very thin and dark, which indicates that these particles lie on their side. This shows that the goethite particles are not entirely cylindrical in shape, but rather form rectangular prisms (similar to the particles in [8]). Goethite grows along the (020) direction [24], shown again in the selected area electron diffraction (SAED) (Figure 3C). Some particles appear to be broken or bent at a 120° angle (Figure 5A). We believe that due to crystal defects and lattice displacements, the (020) direction of the crystals was broken, that the facet grew into two different directions, or that two different particles met randomly, and that their (020) facets grew together. Some evidence for the later theory is shown in high resolution (HR)-TE micrographs of sample G_L (Figure 5B). This is also in accordance with the work of Mao et al. about branched goethite made by oriented aggregation [10].

## 5. Orientation: Results and Discussion

SAXS was used to investigate the changes in orientation when the magnetic field strength is varied. A typical 2D SAXS pattern is shown for sample G_S at a field strength of about 533 mT (Figure 6A). A slight elongation in the x-direction of the image is visible. We measured three samples—G_S, G_M, and G_L—and varied the field between 0 mT and 533 mT. We omitted G_D because of the large fraction of big particles in the sample. From the 2D SAXS pattern we calculated the azimuthal profiles evaluated between *q* = 0.163 nm^−1^ and 0.507 nm^−1^. The azimuthal profiles show only the two peaks at 0° and 180° that develop with rising field strength (Figure 6B). The observed feature has its origin in the short axis of the particles, since the particles orient themselves with their length axis perpendicular to the magnetic field at fields above 70 mT. The length of the particles is not directly visible, due to the accessible *q*-range, this feature is blocked by the beam stop. In Figure 6C we show how the perpendicular orientation to the field develops with rising field strength within an angular range from −90° to 90° for the sample G_S, G_M and G_L. The height of the fit can only be evaluated qualitatively since we did not employ high enough fields to achieve full orientation of the samples, so comparing different samples with different lengths of the particles cannot be done. However, one can observe an increase of the orientation with increasing field strength for each sample.

From the azimuthal profile, we calculated the Herman orientation function $f_{H}$ (Equation (1)) in dependence of the magnetic field (Figure 6D), the required field strength to align perpendicular shifts from 85 mT for sample G_S to 125 mT for the sample G_M and to 147 mT for sample G_L.

The samples in Figure 6 were not concentrated enough to show the parallel orientation above the underground noise. Therefore, we measured another sample G_S*, similar to G_S at 1 vol.% concentration at the BESSY II synchrotron radiation facility (Figure 7). We measured only the field strengths 15, 47, 79 and 615 mT to compare the parallel orientation to the strong orientation. At the three low field strengths, the intensity rings were elongated in the horizontal direction (perpendicular to the field), and at the high field strength the intensity rings were more elongated in the perpendicular direction (parallel to the field), which is slightly visible in the 2D SAXS patterns. This became clearer when the azimuthally averaged intensity was calculated. Here, for better visibility of the orientation of the particles we corrected the data by subtracting the averaged data at 0 mT. Therefore, 0 mT is a straight black line. At 90° the dip in the averaged azimuthal profile stems from the beamstop. Therefore, we analyzed the peak at −90° for the low field strengths. The sample showed almost no orientation at 15 mT, as the Lorentz fit did not converge. At 47 mT and 79 mT, the sample showed parallel orientation and strong perpendicular orientation at 615 mT. The parallel orientation was not as pronounced as the perpendicular orientation. The required field strength to transition to perpendicular orientation should be near the value for G_S at 85 mT, but this sample was measured with a different magnet setup, that had no access to this field strength. The evaluated *q*-range was between 0.125 nm^−1^ and 0.388 nm^−1^.

This behavior can be explained in part by the surface of the particles. Lemaire et al. attributed the parallel orientation to uncoupled magnetic moments on the particle surface [8]. The increase in surface area of the rods from G_S to G_L leads directly to an approximately linear increase in the required field threshold for switching in the observed systems (Figure 6E). As the surface area increases, so does the number of uncoupled magnetic moments that align parallel to a magnetic field.

The behavior can also be understood by employing a micromagnetic model. The total energy of the system is dependent on the angle between the applied field and the magnetization direction of the goethite nanorods. Lemaire et al. concluded that the energy only depends on a Zeeman term, for the interaction between the particle dipole and the applied field, and an induced magnetization by the field which is volume-dependent [8].
(2)$$E\left(\theta \right)=-\mu Bcos\theta -\frac{\Delta \chi V}{2\mu_{0}}B^{2}cos^{2}\theta$$

Here μ is the magnetic moment of the particles, θ is the angle between the field and the magnetization direction of the particles and Δχ is the magnetic susceptibility anisotropy. The coefficients μ and $\Delta \chi V/2\mu_{0}$ are of the same order of magnitude, so the two terms are mainly dependent on B and $B^{2}$. At low field strengths, the Zeeman term dominates, aligning the particles parallel to the field. The energetic minimum is here at 0°. At higher fields, the $B^{2}$ term of the induced magnetization dominates, and the negative anisotropy of the magnetic susceptibility aligns the particles perpendicular to the field with an energetic minimum at 90°. The crossover was determined to be at $B=\frac{\mu \mu_{0}}{\left|\Delta \;\chi \right|V}$ with values of 100 mT [8], but we observe lower values around 100 mT, and the crossover is more dependent on the surface than on the volume. 

## 6. Conclusions

In this work we presented a facile hydrothermal synthesis of goethite nanorods with a tunable aspect ratio. The rods show good crystallinity and low width dispersity. We could show that the particle surface has a distinct influence on the field strength necessary to align the goethite nanorods. The particles align parallel at low fields and perpendicular at high fields. However, the parallel orientation is not visible if the concentration of the particles is low due to the accessible *q*-range and intensity. The required field strength to reorient the particles to align perpendicularly increases from 85 mT for the short (210 nm) rods (G_S) to 147 mT for the long (320 nm) nanorods (G_L). This is in line with the existing literature by Lemaire et al., which attributes the parallel orientational behavior to the unaligned surface moments of goethite that can follow the magnetic field director [8]. The behavior can be explained by a micromagnetic model, but we observed some differences compared to the previously documented results. We believe that the anisotropic properties of goethite will show very interesting results when combined with a matrix. A polymer/nanorod composite with switchable anisotropy could have potential applications for membranes or sensor technology. 

## Figures and Tables

**Figure 1 nanomaterials-10-02526-f001:**
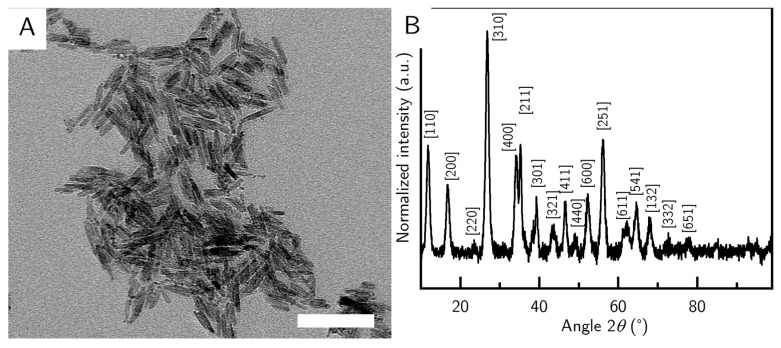
(**A**) TEM image of the akaganeite precursor particles. (**B**) XRD of a precursor sample. The references were matched to an akaganeite reference.

**Figure 2 nanomaterials-10-02526-f002:**
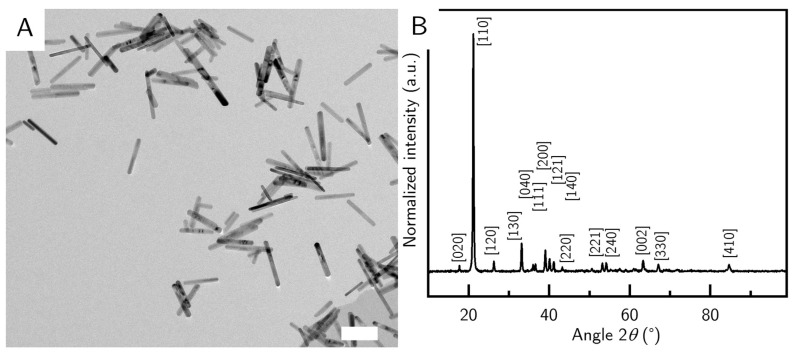
(**A**): TEM image of sample G_S. The scale bar shows 200 nm. (**B**): XRD of sample G_S. The Miller indices (HKL values) are from the goethite reference.

**Figure 3 nanomaterials-10-02526-f003:**
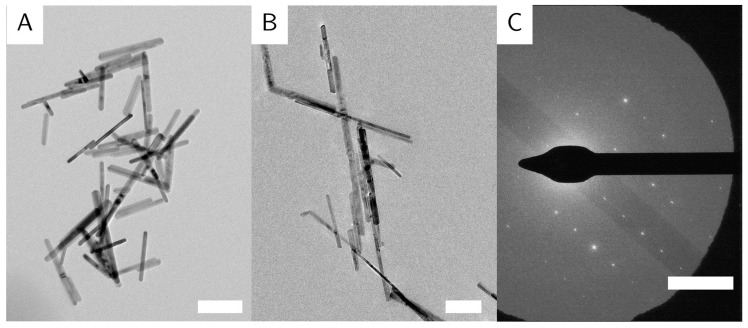
TEM images of selected goethite particles from Table 1. (**A**): G_M. (**B**): G_L. (**C**) Selected area electron diffraction (SAED) overlaid over a particle from sample G_L displayed in (**B**). The scale bar shows 200 nm for the TE micrographs and 5 nm^−1^ for the SAED.

**Figure 4 nanomaterials-10-02526-f004:**
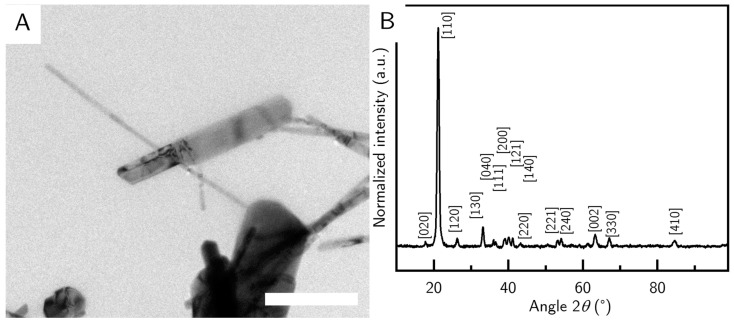
(**A**) TEM image of sample G_D. The scale bar depicts 500 nm. (**B**) XRD of sample G_D. The peaks were referenced according to the goethite reference.

**Figure 5 nanomaterials-10-02526-f005:**
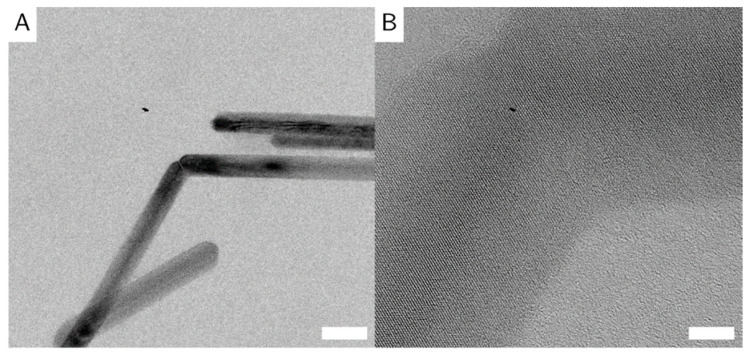
HR-TE micrographs of interconnected goethite nanorods of sample G_L. On the left (**A**), zoom of the junction between the particles on the right (**B**). The scale bars show 50 nm (**A**) and 5 nm (**B**).

**Figure 6 nanomaterials-10-02526-f006:**
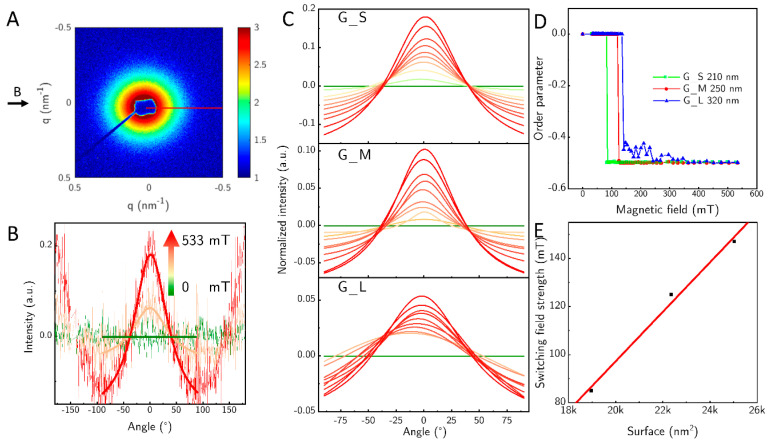
Orientation of the goethite rods in a magnetic field with rising field strength measured by SAXS. (**A**) 2D image of an example SAXS scattering experiment, where, G_S is at 533 mT. The magnetic field was applied by permanent magnets from the side of the sample. The red line is depicting 0° for the azimuthal plots. (**B**) Azimuthal profile of the measured intensity integrated along each angle. The measured field strengths were 33 mT (green), 153 mT (orange), 533 mT (red). The thin lines show the measured intensity, while the strong lines indicate the resulting fits. (**C**) Fitted intensity perpendicular to the magnetic field and to the beam for the three pure samples. The field strength is depicted by the arrow in the middle from 0 to 533 mT (**D**) The Herman orientation function *f_H_* in dependence of the field. (**E**) Required field strength for the switching of the particles versus the surface area of the particles.

**Figure 7 nanomaterials-10-02526-f007:**
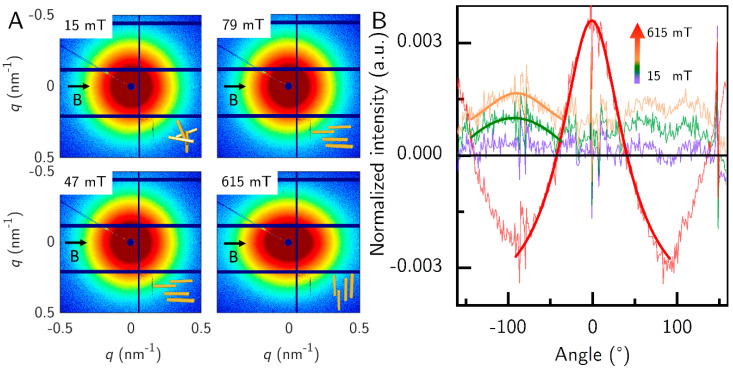
Orientation of a sample G_S* at 1 vol.% particle concentration in a strong field. (**A**) 2D images with the respective orientation of the goethite nanorods are shown for each measured field strength. (**B**) Azimuthal profile of the normalized intensity measured along each angle, shifted in intensity for clarity. The solid lines depict Lorentz fits, while the thin lines depict the normalized intensity extracted from the 2D images in (**A**).

**Table 1 nanomaterials-10-02526-t001:** Conditions for each synthesis: Addition sequence 1, PVP is added prior to the precursor; addition sequence 2; the precursor is added prior to PVP. Dialysis depicts whether the precursor was dialyzed to remove excess NaH_2_PO_4_, the reaction time is the time during which the temperature was maintained at 160 °C. For the stirring, different stirring bars are used: A small stirring bar is 12.3 mm in length and 4.5 mm thick, and a big stirring bar is 20.3 mm in length and 6 mm thick.

Sample Name	Addition Sequence	Dialysis	Reaction Time (h)	Stirring Bar	Stirring (rpm)
G_S	1	-	2	Small	100
G_S*	1	-	2	Small	100
G_M	1	-	2	Big	100
G_L	2	-	2	Small	100
G_D	1	+	12	Small	100
G_noPVP		-	2	Small	500

**Table 2 nanomaterials-10-02526-t002:** Selection of particles that were investigated by SAXS. The length and width of the particles were determined by TEM, the aspect ratio is Length/Width and the switching field strength is the minimum field strength required to switch the orientation from parallel to perpendicular to the field.

Sample Name	Length (nm)	Width (nm)	Aspect Ratio	Switching Field Strength (mT)
G_S	210 ± 61	27 ± 9	8	85
G_M	250 ± 77	27 ± 8	9	125
G_L	320 ± 97	24 ± 7	13	147
G_S*	204 ± 62	30 ± 10	7	Not measured
G_D	650 ± 330	32 ± 13	20	Not measured
G_noPVP	218 ± 70	33 ± 11	7	Not measured

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
