# Peer review of "Goethite Nanorods: Synthesis and Investigation of the Size Effect on Their Orientation within a Magnetic Field by SAXS"

_nanomaterials, 2020, doi:10.3390/nano10122526_

Round 1

Reviewer 1 Report

The paper by Hinrichs et al. is devoted to the exploration of the behavior of goethite-type ferrite nanorod assemblies, in external magnetic field, in a mode analogous to that in nematic/liquid crystal displays. The paper manuscript contains well covered literature background, detailed experimental section and comprehensively written results section. Limitations of the measurements related to the too low concentration of the rods in a medium to be possibly created to develop a sensor technology was also recognized, eventually. The only and very minor (nowadays) deficiency is that some essential parts of the illustrations to the results, that is figures 6 and 7 depicting the SAXS-evaluated orientation against field or with field as the parameter, would not be readable in grayscale print.

Last, but not least, the topic of the paper is, indeed, interesting, besides its well arranged presentation. The manuscript may well become published in the present form.

Author Response

Thank you for your feedback. Regarding the grayscale print, we believe that this is unfortunate but also think that, with the amount of information contained in the colorscale, a grayscale image would become too cluttered to be easily comprehensible.

Reviewer 2 Report

I appreciate the work done for this paper. I consider that the team that worked on this article has done a great job for the scientific world that is concerned with the field of materials.

Author Response

Thank you for your feedback.

Reviewer 3 Report

The work shows a nice investigation on the characterization and orientation on gothite nanorods by combinig three main techniques: XRD and TEM for the structural characterization of the samples and SAXS for the study of the orientation induced by an external magnetic field. I found the work interesting and well structured.

The first part of the manuscript referring to the structural characterization by combinig TEM and XRD, is well detailed and comprehensive.

In the line 196 I don't really get the meanig of the sentence '.. the reaction is easy to scale up', for me is not clear how the author get this conclusion. Please explain or remove it.

The second part of the manuscript is interesting but in my opinion needs to be more comprehensive, I suggest to add one or two 2D SAXS images where is more evident the orientation, this will help also the reader to understand what is done and the techique

There are come typos please check

Author Response

Thank you for your fruitful comments.

We changed the sentence in line 196 by splitting the sentence after the first part and changed the second part

“This means, by increasing the precursor concentration, the reaction can be easily scaled up, leading to a higher yield.“

We added four more SAXS images in figure 7 and changed the caption accordingly to be more comprehensive. In the text we added several sentences in chapter 5

“However, one can observe an increase of the orientation with increasing field strength for each sample.“

In figure 7: “. (A) 2D images with the respective orientation of the goethite nanorods are shown for each measured field strength. (B) Azimuthal profile of the normalized intensity measured along each angle, shifted in intensity for clarity. The solid lines depict Lorentz fits while the thin lines depict the normalized intensity extracted from the 2D images in (A).”

“At the three low field strengths the intensity rings are elongated in horizontal direction (perpendicular to the field) and at the high field strength the intensity rings are more elongated in the perpendicular direction (parallel to the field), which is slightly visible in the 2D SAXS pattern. This becomes clearer, when the azimuthally averaged intensity is calculated. Here, for …“

We checked for typos again.